# Controlling Calcium Carbonate Particle Morphology, Size, and Molecular Order Using Silicate

**DOI:** 10.3390/ma14133525

**Published:** 2021-06-24

**Authors:** Lior Minkowicz, Arie Dagan, Vladimir Uvarov, Ofra Benny

**Affiliations:** 1The Institute for Drug Research, The School of Pharmacy, Faculty of Medicine, The Hebrew University of Jerusalem, Jerusalem 91120, Israel; lior.minkowicz@mail.huji.ac.il (L.M.); arie.dagan@mail.huji.ac.il (A.D.); 2The Center for Nanoscience and Nanotechnology, The Faculty of Science, The Hebrew University of Jerusalem, Jerusalem 91120, Israel; vladimiru@savion.huji.ac.il

**Keywords:** calcium carbonate, sodium silicate, calcite, EDS, nanoparticles

## Abstract

Calcium carbonate (CaCO_3_) is one of the most abundant substances on earth and has a large array of industrial applications. Considerable research has been conducted in an effort to synthesize calcium carbonate microparticles with controllable and specific morphologies and sizes. CaCO_3_ produced by a precipitation reaction of calcium nitrate and sodium carbonate solution was found to have high polymorphism and batch to batch variability. In this study, we investigated the polymorphism of the precipitated material and analyzed the chemical composition, particle morphology, and crystalline state revealing that the presence of silicon atoms in the precipitant is a key factor effecting particle shape and crystal state. An elemental analysis of single particles within a polymorphic sample, using energy-dispersive X-ray spectroscopy (EDS) conjugated microscopy, showed that only spherical particles, but not irregular shaped one, contained traces of silicon atoms. In agreement, silicon-containing additives lead to homogenous, amorphous nanosphere particles, verified by X-ray powder diffraction (XRD). Our findings provide important insights into the mechanism of calcium carbonate synthesis, as well as introducing a method to control the precipitants at the micro-scale for many diverse applications.

## 1. Introduction

Calcium carbonate (CaCO_3_) is one of the most abundant substances on earth. This mineral is found naturally in rocks and is the main component of pearls and of shells of marine organisms. Calcium carbonate has three main crystalline states, calcite, vaterite, and aragonite, which each having typical morphology rhombohedral, spherical, and needle-like, respectively [1]. This mineral has many utilizations; it is a main active ingredient in agricultural lime [2], it is being used as a filler material in the paper and colors industries, and is widely used as building material in the construction industry [3,4]. In medicine, it is used as an inexpensive dietary calcium supplement [5,6], an anti-acid, [7], and an inert filler for tablets and other pharmaceuticals [8]. Due to its positive charge and erosion capability, calcium carbonate has also been explored preclinically as a template for layer-by-layer (LbL) deposition in drug delivery applications [9,10,11]. As with any particulate pharmaceutical product, the ability to control the physicochemical properties of the product is a critical issue as particle size, shape, and composition dictate the drug’s performance, e.g., biodistribution, selectivity, and elimination from the body. Despite remarkable advantages for calcium carbonate, including low cost and high biocompatibility, its utilization in biomedicine is limited by the low reproducibility of the synthesized end-product and high polymorphism that make the control over particle morphology very challenging. Therefore, a key task in this regard would be to identify which factors affect the synthesis and to correlate between material composition and morphology, thus providing a predictable product with well-defined properties.

In this study, we attempted first to evaluate the reproducibility of the synthesis reaction and identify what components affect the morphology of the end-product. We therefore performed microscopical morphology study followed by chemical analysis in order to identify these potential effectors. The calcium carbonate particles were obtained by precipitation reaction using soluble calcium nitrate and sodium carbonate as previously described [1,12,13] with or without the addition of carboxymethylcellulose (CMC) [12], which was previously shown to affect particle shape [14]. Surprisingly, we found that despite repeating the exact same protocol, the product was still highly polymorphic and had wide batch-to-batch variability. By analyzing the product using energy-dispersive X-ray spectroscopy (EDS) we revealed that only particles carrying spherical shape contained residual silicon atoms (<1%), while non-spherical particles had no detectable silicon atoms. To further validate our findings, we repeated the original precipitation reaction in the presence of silicon-containing additives added exogenously. Different silicate additives were explored, revealing that the presence of silicate shifts the synthesis towards the production of spherical nanoparticles in a concentration-dependent manner. Moreover, we found that the contained silicate preserved the amorphous state of the mineral, while in its absence, the mineral was in its crystalline, calcite, state. Various concentrations of soluble silicate produced nanospheres with range of silicon atom content, measured by inductively-coupled plasma optical emission spectroscopy (ICP-OES). Further studies are required to determine the exact mechanism of crystallization and the role of silicate in this reaction. The ramifications of the current work are the possibility of being able to better control the morphology of silicon-containing calcium carbonate composite particles and as a result increasing its use in biomedicine in the form of nanoparticles. 

## 2. Experimental Methodology

### 2.1. Preparation of Calcium Carbonate (CaCO_3_) Particles

Calcium carbonate particles were formed by a chemical reaction between CO_3_^2−^ and Ca^2+^ as described elsewhere [12]. The following chemical reaction was performed, with or without CMC (Formula (1)):Na_2_CO_3_ + Ca(NO_3_)_2_ → CaCO_3 (s)_ + 2Na^+^ + 2NO_3_^−^(1)

The basic protocol used throughout the study is detailed below and any variation is mentioned specifically in the text. Sodium carbonate solution (0.025 M; Sigma, S2127, St. Louis, MO, USA) was mixed with 0.17% *w*/*v* CMC (Fluka, BioChemika, 21901, ultra-low viscosity, 20–50 mPa·s) in a volume of 25 mL DDW and sonicated for 1 min. A solution of 0.025 M (25 mL) calcium nitrate in DDW (Sigma, Steinheim, Germany #31218) was added and the CaCO_3_ was allowed to precipitate under a magnetic stirrer for 5 min. The product was then centrifuged for 5 min at 1000× *g*. The precipitate was washed twice with DDW and lyophilized for 48 h. Sodium silicate solution in a density of 1.4 g/mL (Sigma, St. Louis, MO, USA, #338443) was added to the sodium carbonate solution or, alternatively, was mixed together with the calcium nitrate.

### 2.2. Characterization of Crystal State of CaCO_3_ Particles by Powder X-ray Diffraction

The crystalline structure of the specimens was analyzed by XRD. Measurements were performed using a D8 Advance diffractometer (Bruker AXS, Karlsruhe, Germany). A low background quartz sample holder was carefully filled with the powder samples. The XRD patterns from 5 to 70° 2θ were recorded at room temperature using CuKa radiation (λ = 0.15418 nm) with the following measurement conditions: Tube voltage of 40 kV, tube current of 40 mA, step scan mode with a step size 0.02° 2θ, and counting time of 1 s per step. Phase identification and quantification were performed using EVA software (Bruker AXS). 

### 2.3. Morphological Studies Using Electron Microscopy 

Morphology was observed and chemical composition was identified with the environmental Scanning Electron Microscope (SEM) Quanta 200 (FEI Company, The Netherlands), equipped with an energy-dispersive X-ray spectroscopy detector (EDAX-TSL, AMETEK, Hillsboro, OR, USA). For SEM analysis and imaging, the particles were sampled on a conductive adhesive tape, metal coated with a thin film of Pd/Au sputtered onto the sample (SC7620 Sputter coater, East Sussex, UK) and air-dried. Some of the samples were subjected to chemical elemental analysis with an EDS function. The elemental composition of a sample was determined using a characteristic X-ray spectrum. The analysis was performed either in a frame mode, which sums the elemental composition of the particles, or in a spot mode, in which the beam targets a single area chosen within the field of view that facilitates the characterization of various particles in a single sample.

Transmission electron microscopy (TEM) observations were carried out with Jeol, TEM 1400Plus, Japan, with a charge-coupled camera Gatan Orius SC600 to detect nanoparticles. Particles were stained with 2% uranyl acetate; the observation was subsequently made in a carbon film.

### 2.4. Inductively-Coupled Plasma Optical Emission Spectroscopy

ICP-OES quantitative elemental analysis was performed using PQ 9000 Elite (Analitik, Jena, Germany). Calibration was done with MERCK solution and accuracy confirmations were performed according to the standards of the US Geological Survey (USGS). Powder formulations were dissolved in 50 mL DDW in 1% nitric acid (HNO_3_) before elemental analysis.

## 3. Results

### 3.1. Analyzing Morphology and Composition of Calcium Carbonate Samples

Production of calcium carbonate particles in a reproducible manner and with a predictable shape is highly challenging and standard protocols would often yield high sample variability. Elucidating the parameters that affecting calcium carbonate particles can greatly contribute to the establishment of robust synthesis protocols. Producing uniform products at the micro or nanometric scales are conditional step toward expanding the utilization of calcium carbonate particles in the high standard required in biomedicine.

To assess calcium carbonate polymorphism and determine the variability of the reactions we first followed standard protocols using sodium carbonate, CMC, and calcium nitrate [12]. Repeating the exact procedure multiple times revealed high batch to batch variability as shown with SEM imaging in Figure 1 (representative images of three independent batches). In each batch particles acquired entirely different morphologies of spheres, cubes, and ellipsoids. We attempted to control particle formation by erosion by adding calcium chelator (0.25 µM ethylenediaminetetraacetic acid (EDTA)) to the synthesis reaction for various durations (2, 5, and 15 min). However, there was no advantage over product uniformity and the sample contained diverse product with size range of ~1–5 µm (Appendix A).

In order to determine the origin of the polymorphism in samples, we performed a composition EDS/SEM analysis for spherical particles, in comparison to the non-spherical crystals at the same sample. As shown in Figure 2 and Appendix A, we consistently found that only spheres, and not other shapes, contained a very low (<1%) amount of silicon atoms in addition to the other expected elements. The presence of gold traces was the result of sample coating prior to SEM analysis. Based on this unexpected observation, we attempted to determine whether silicon-containing additives may shift the morphology of the calcium carbonate towards specific spherical morphology. Therefore, we repeated the reaction in the presence of various silicon-containing additives.

### 3.2. Addition of Silicon-Containing Additives and their Effect on Calcium Carbonate Particles

Commercial additives—silicon grease and silicone gel—that contain a mixture of silicate molecules in an undefined composition were dispersed or solubilized in organic solvents and were added to the precipitation reaction of calcium carbonate, as shown in (Appendix A). In the presence of silicone grease dissolved in ethyl acetate (EtOAc), small, uniform microparticles were obtained (Appendix A). Analysis with EDS showed, in agreement to our initial observation, that traces of silicon atoms were present in spheres but not in cubes. Appendix A summarizes the morphology data obtained with the various silicon-containing additives.

In order to study the effect of specific and defined silicon-containing molecules on calcium carbonate synthesis, we selected a soluble form of silicon compound, i.e., sodium silicate. Commercial sodium silicate solution at a concentration of 1% *v/v* was added to the synthesis, and particle morphology was studied (Figure 3). In the presence of silicate, the particles were smaller than previously obtained with nanoparticles size range of 30 to 50 nm with larger particles of ~200–250 nm (Figure 3B). 

### 3.3. Crystalline State of Calcium Carbonate with Various Content of Silicate 

The crystalline state of the material was determined by X-ray powder diffraction (XRD). 

The results showed that silicate containing products maintained their amorphous state of the material, while calcite particles were formed in the absent of silicate (Figure 4A,B). Therewith, calcite was found in pure form when particles were synthesized without sodium silicate, and the mean crystallite size of calcite calculated using the Scherrer equation [15] was ~40 nm. To eliminate the possibility that the basic pH of the silicate solvent affected particle state rather than the silicate itself, we repeated the reaction in the presence of 0.1% *v/v* NaOH in water at the same concentrations as in the silicate solution. As can be seen in Figure 4C, we confirmed that pH had no effect on the morphology of calcium carbonate particles. 

Importantly, the crystal state of the material was verified with XRD for varying silicate contents (Figure 5A). It was found to be a mixture of calcite and vaterite with approximate ratio 20:80. The crystal size of calcite and vaterite calculated using the Scherrer equation was 87 nm and 22 nm, respectively. The crystal state of the material was verified with XRD for varying silicate contents and quantitative analysis of samples with increasing amounts of sodium silicate was performed using ICP-OES. The analysis showed that 0.05 and 0.1 mL of sodium silicate resulted in atomic ratios of ~1:4 and ~1:2 for silicon:calcium (Figure 5B). 

### 3.4. The Effect of CMC and Product Stability

Repeating the synthesis in the presence of silicate but without CMC yielded similar results to the CMC-containing samples, suggesting that CMC does not have a critical role in the formation of nanospheres (Figure 6). Importantly, the material was found to be stable over two weeks without apparent visual changes (Figure 7A), and the reaction was unaffected by high temperature as both 4 °C and 45 °C reactions produced similar products in terms of size and morphology (Figure 7B).

## 4. Discussion

Calcium carbonate particles are widely used in various industrial and biotechnological applications due to their beneficial properties such as high porosity, non-toxicity, and biocompatibility [8,16,17]. This polymorphic compound has various structural states, of which the amorphous calcium carbonate is relatively short-lived and unstable and is thought to act as a seed for the crystal growth of the other three polymorph crystals. These crystals, namely, calcite, vaterite, and aragonite, have, respectively, rhombohedral, spherical, and needle-like morphologies [1]. The transition between the different states is a topic of much research. Several reports concerning calcium carbonate have described spheres, cubes, and other shapes, such as ellipsoids and discs [1,12,13]. However, in most cases, the reproducibility of the synthesis, as well as the key reaction parameters that govern the shape of the particles, were not reported. 

Many efforts to improve reproducibility and control the calcium carbonate properties included the addition of various excipients to the precipitation reaction. When used as a pigment material, the addition of CMC was shown to affect calcium carbonate particle morphology, improve optical performance, and reduce particle mechanical strength, which can be further improved by supplementation with corn starch [18]. In another study, CMC was shown to induce spherical particle formation [14] and contribute negative charges, thus making the particles suitable as templates for LbL deposition [12]. In our experiments, using CMC as a single additive revealed that the presence of silicate was the critical factor for producing nanospheres and not the CMC itself as similar nanospheres were obtained with or without the CMC as shown in Figure 6. Additional studies used additives that affected particle morphology and size. For example, Yu et al. showed that the addition of polyacrylic acid induced the formation of cubic monodispersed calcite particles, emphasizing the importance of a single additive in the synthesis process on particle morphology [19]. Similarly, it was shown that EDTA triggers the formation of spherical particles, while citrate shifts the reaction to form mainly rhombic particles [20]. In our study, we could not detect a clear effect of EDTA on the product.

In drug delivery, nano- and microparticles are often being utilized as carriers with superior pharmacological benefits comparing to free drugs [21,22]. In this respect, calcium carbonate may be useful due to its biocompatibility and low cost. Particle biophysical properties affect drug biodistribution, selective targeting, and elimination from the body [23,24,25]. Therefore, extensive efforts have been invested in controlling the fabrication of particles, either as drug carriers or as templates for LbL formulations. The ability to reproduce the production of calcium carbonate was determined by repeating the precipitation reaction for several times and characterize the end-product. The results indicated that the product is highly polymorphic and carries wide batch-to-batch variability. Analysis with EDS/SEM, enabled to detect traces of the element silicon but only in the spherical particles. This observation strongly suggests that the presence of silicon affects the production of spherical shaped particles. One of the proposed explanations is that silicon stabilizes the amorphous state of the material, as has been suggested in regard to other molecules. Aizenberg et al. [26] showed that the presence of magnesium and macromolecules extracted from different biogenic sources shifted the synthesis of calcium carbonate from calcite to amorphous calcium carbonate and/or inhibited crystallization in vitro. In agreement with these observations, we found that silicate shifted the synthesis of calcium carbonate to an amorphous state.

Many previous reports that described the production of spherical calcium carbonate might have also contained unintentional silicon traces, possibly due to contamination from glassware or other sources. To establish a robust protocol that can be repeated by others, silicon must be supplemented exogenously to the reaction in a controlled manner. Therefore, we used different sources of silicon-containing materials, some insoluble (silica gel and grease) and some soluble (dissolved grease in ethyl acetate/sodium silicate solution). Out of all additives tested, we found that dispersed silica grease in ethyl acetate produced the most uniform microspheres, and interestingly, the addition of soluble sodium silicate produced smaller particles at the nano range. Further analysis of the thermostatic state of the nanospheres revealed that the material remained in its amorphous state and did not crystallize. We assume that the presence of silicon in the sample might stabilize the amorphic state of the material as the crystal begins to form at the submicroscopic level, possibly acting partly as an antiscalant for calcium carbonate, ref. [27] inhibiting the formation of the crystallized mineral and maintaining it at a nanoscale and in an amorphic state.

Quantitative analysis of the nanospheres (Figure 5) showed that doubling the sodium silicate content increased the silicon:calcium ratio in the particles (from ~1:2 to ~1:4).

## 5. Conclusions

Precipitation of calcium carbonate is an environmental sensitive reaction with delicate process that reduce the capacity to control the end product. There is a high innate polymorphism in a given sample as well as high batch to batch difference. Our study revealed that the present of silicon in the precipitants affect the morphology of the mineral thus producing spherical and amorphous material. The synthesis of calcium carbonate was highly affected by to the presence of silicates residues and the exact composition of the additive affect the nature of the end product in terms of shape and size. Finally, we concluded that it is possible to control the morphology of calcium carbonate and it size where a minimal ratio of 1:4 silicon: Calcium is required to yield nanosphere-shaped particles. The capacity to control the production of the material is expected to impact substantially its potential usage in various fields including biomedicine. 

## Figures and Tables

**Figure 1 materials-14-03525-f001:**
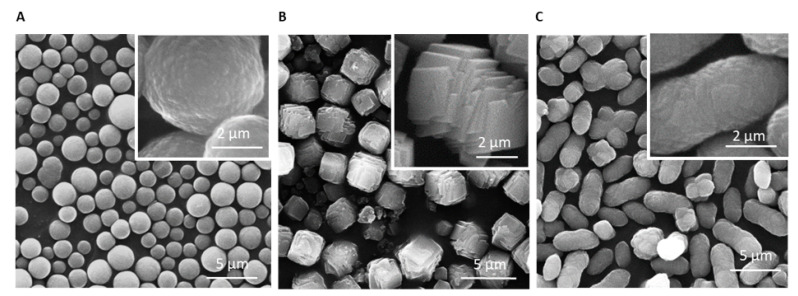
Calcium carbonate fabricated with CMC produces polymorphic samples and heterogenic batches. SEM images (**A**–**C**) from three independent samples, fabricated with the same standard protocol, show high batch to batch variation. Magnification ×10,000.

**Figure 2 materials-14-03525-f002:**
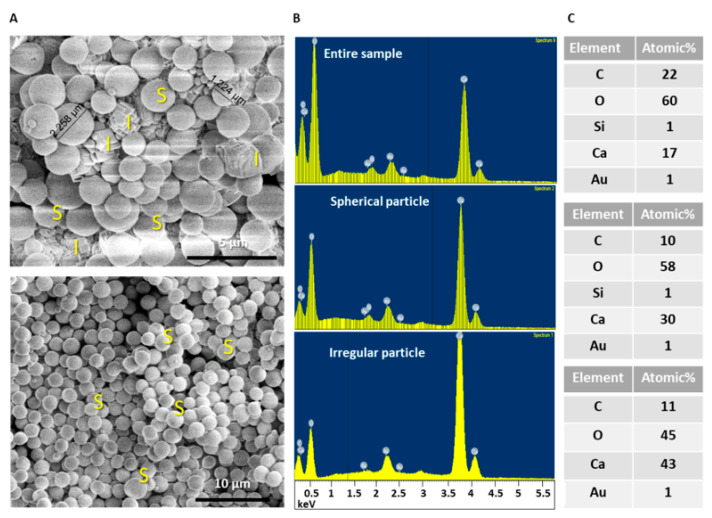
Silicon element found in sphere particles after EDS elemental analysis of CaCO_3_ with CMC (sample 1A). The analysis includes the overall sample, single spheres, and single irregular shaped particles. (**A**) SEM images of the particles which were targeted by EDS. (**B**,**C**) Elemental analysis of the overall sample and targeted particles. Silicon was found in the entire sample and in spherical particles.

**Figure 3 materials-14-03525-f003:**
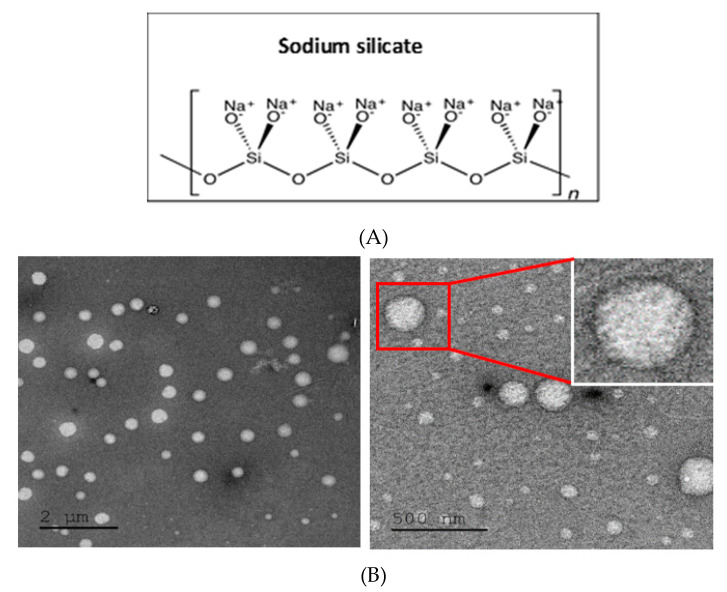
Effect of sodium silicate on particle size and shape. (**A**) Chemical structure of sodium silicate. (**B**) TEM images of particles fabricated with sodium silicate and CMC showing nanospheres with a size range of 30 to 50 nm together with some larger particles ~200–250 nm.

**Figure 4 materials-14-03525-f004:**
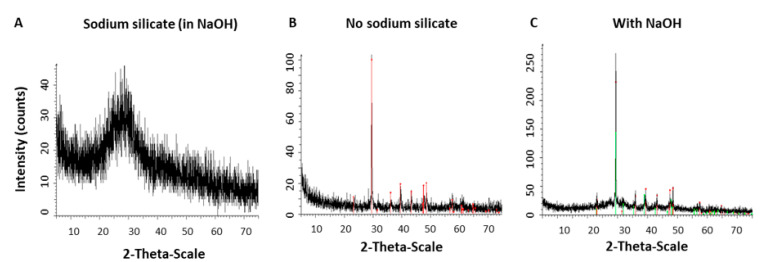
Effect of sodium silicate on the crystal state of the material. (**A**,**B**) XRD analysis of the particles synthesized with sodium silicate resulted in an amorphous material, while absence of silicate produced the high ordered form of the material, calcite. (**C**) Control, only the solvent of silicate solution was added (NaOH) showing no effect on the material’s crystalline state.

**Figure 5 materials-14-03525-f005:**
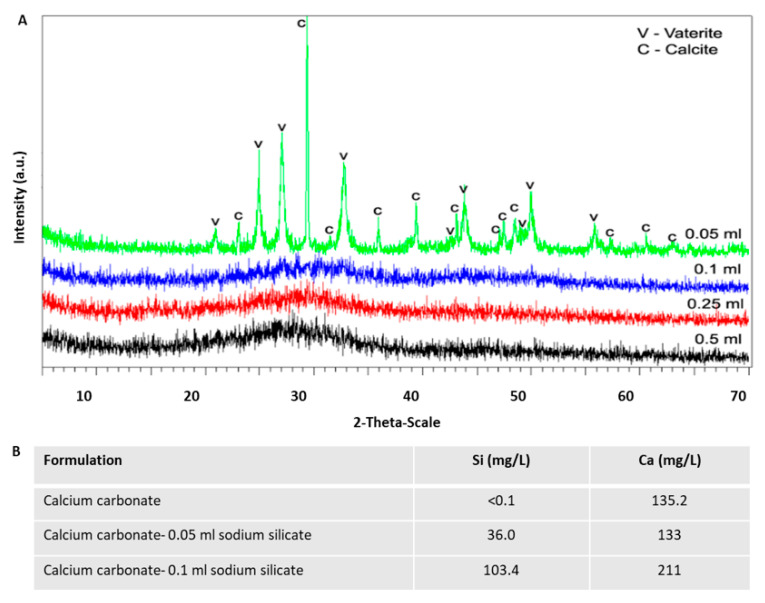
Calcium carbonate synthesized with elevated concentrations of sodium silicate. Decreasing sodium silicate concentration from 0.5 mL (1% *v/v*) to 0.05 mL resulted in a shift from amorphous material (high silicate) to calcite (no silicate) as assessed by XRD graphs (**A**) and numeral data (**B**).

**Figure 6 materials-14-03525-f006:**
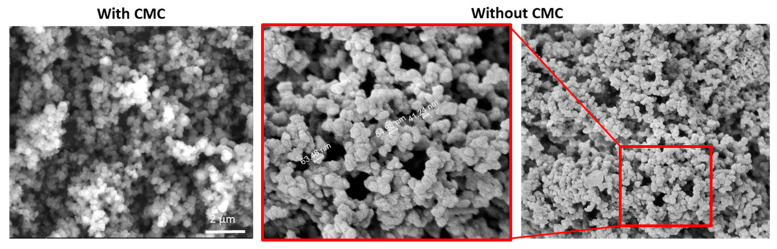
High-resolution SEM images of calcium carbonate particles produced with and without CMC. Nanospheres were obtained with the presence of silicate. Similar morphologies were obtained with or without the presence of CMC.

**Figure 7 materials-14-03525-f007:**
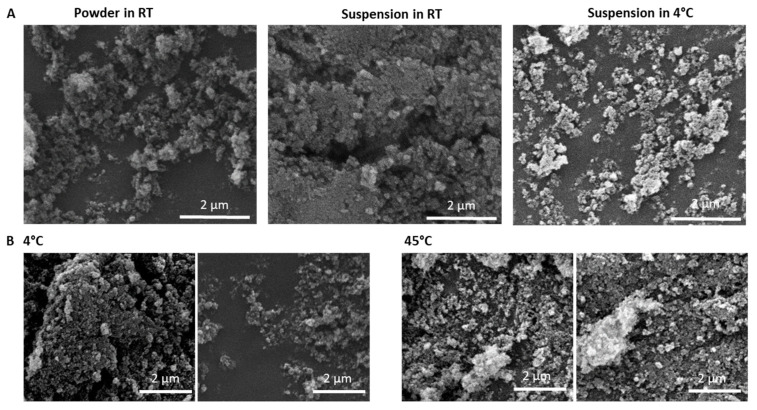
Stability of Calcium carbonate-silicate in different storage conditions and temperature. (**A**) SEM images of samples that were stored as a powder in room temperature; suspension in room temperature; or as a suspension in 2–8 °C. (**B**) SEM images of synthesized in 4 °C or in 45 °C.

## Data Availability

Data sharing not applicable.

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
