# Peer review of "Controlling Calcium Carbonate Particle Morphology, Size, and Molecular Order Using Silicate"

_materials, 2021, doi:10.3390/ma14133525_

Round 1

Reviewer 1 Report

In this study, the researcher investigated the polymorphism of the precipitated material and were able to correlate chemical composition, particle-morphology, and crystalline state. 

  1. It seems there is no conclusion in this paper.
  2. It is very highly advised to cite: 10.1016/j.pnsc.2020.12.004 10.1016/j.jmapro.2020.11.043  in related areas.
  3. Can you add the XRD data to confirm the property of your particles?
  4. Can you add the mechanism of your research?
  5. It seems this research is not complete, but others are okay.

Author Response

In this study, the researcher investigated the polymorphism of the precipitated material and were able to correlate chemical composition, particle-morphology, and crystalline state.

  1. It seems there is no conclusion in this paper.

Au: A conclusion section was added.

  1. It is very highly advised to cite: 10.1016/j.pnsc.2020.12.004 10.1016/j.jmapro.2020.11.043 in related areas.

Au: The references mentioned discuss Cu-Cu joins which are not directly relevant to the paper topic, we could not find the context they should be referenced.

  1. Can you add the XRD data to confirm the property of your particles?

Au: XRD Data are provided in figures 4 and 5.

  1. Can you add the mechanism of your research?

Au: The exact mechanism is yet to be resolved but we provided a correlative information related to the link between composition to structure using systematic approaches.Moreover, we provided the information related to the minimal silicon:calcium ratio required to obtain amorphous material.

  1. It seems this research is not complete, but others are okay.

Au: We are not entirely sure the comment. We have provided several studies confirming the correlation of silicate content to the morphology of the CaCO3 obtained. The information is novel and significant for many potential applications. We re-written many parts of the introduction and discussion and we hope the reviewer will find the text flow improved.

Reviewer 2 Report

Authors of the article entitled 'Controlling Calcium Carbonate Particle Morphology, Size and Molecular Order Using Silicate' showed very interesting results on the fact, how silicate can influence the production of calcium carbonate particles. However, they did not avoid many mistakes, so the article cannot be published in its current form. Mistakes are as follows:

- Abstract- should be more concise; authors may, for example, add a sentence regarding the preparation of calcium carbonate particles;

- Introduction- the second paragraph is practically 'conclusions', it should be rewritten; there is no paragraph where other attempts to adjust the control of morphology and particle size of calcium carbonate would be described; you may here rely, for example, on the starch; there is also no explanation of what the new described research brings to the development of science in this area, and there is no comparison of the described data with other results available in the literature (or an explanation that this is pioneering research in this field);

- chapter 4 ('Methods') should be moved to the beginning, right after the 'Introduction', and it should have a different name, for example 'Experimental section'- 'Preparation of CaCO3 particles' should be a subsection 2.1, and subsection 4.1 should be a subsection 2.2 etc.;

- subsection 'Preparation of CaCO3 particles'- chemical reaction should have a number; labels with the size of the packaging should be removed; unit at viscosity - the dot should be in the middle, not at the bottom; the statement 'in short' should be removed; designation of gravity- it should be a dot instead of coma and 0 should be the same size (or just write 1 g); silicate modulus of the sodium silicate solution should be given;

- chapter 3, 'Discussion'- this chapter should be moved to the 'Introduction' section; this chapter should be about discussing the reported results; only the last paragraph is suitable for the 'Discussion';

- there is no 'Summary' or 'Conclusions' chapter in this article;

- Figure captions in the main text should be short and refer only to what is shown in the specific figure;

- Results- this chapter should be divided into two subsections, i.e. I- concerns only the preparation of CaCO3 particles samples, II- concerns the influence of the silicon-containing additives;

- Figure 1S- should be in the main text, not in the supplementary material;

- p.2- lines 81-86- there is no reference to this fragment in the subsection 'preparation ..', where only the sodium silicate solution is mentioned; what exactly were these gels?; Figure 3S should be in the main text, not in the supplementary material; Figure 7- wrong numbering, please correct it;

- Figure 3- the marks are not well visible on it, descriptions have been 'cutted'; please remove fragment A from the figure - it does not add anything significant to the text;

- Figures 5S and 6S- should be in the main text, not in the supplementary material.

Author Response

Authors of the article entitled 'Controlling Calcium Carbonate Particle Morphology, Size and Molecular Order Using Silicate' showed very interesting results on the fact, how silicate can influence the production of calcium carbonate particles. However, they did not avoid many mistakes, so the article cannot be published in its current form. Mistakes are as follows:

- Abstract- should be more concise; authors may, for example, add a sentence regarding the preparation of calcium carbonate particles;

Au: We thank the reviewer for the comment. We made the required corrections.

- Introduction- the second paragraph is practically 'conclusions', it should be rewritten; there is no paragraph where other attempts to adjust the control of morphology and particle size of calcium carbonate would be described; you may here rely, for example, on the starch; there is also no explanation of what the new described research brings to the development of science in this area, and there is no comparison of the described data with other results available in the literature (or an explanation that this is pioneering research in this field);

Au:. Per reviewer’s comment we re-written the introduction and re-order it with discussion in order to clarify better the gap and our contribution to the current knowledge.

- chapter 4 ('Methods') should be moved to the beginning, right after the 'Introduction', and it should have a different name, for example 'Experimental section'- 'Preparation of CaCO3 particles' should be a subsection 2.1, and subsection 4.1 should be a subsection 2.2 etc.;

Au: We have modified accordingly.

- subsection 'Preparation of CaCO3 particles'- chemical reaction should have a number; labels with the size of the packaging should be removed; unit at viscosity - the dot should be in the middle, not at the bottom; the statement 'in short' should be removed; designation of gravity- it should be a dot instead of coma and 0 should be the same size (or just write 1 g); silicate modulus of the sodium silicate solution should be given;

Au: We thank the reviewer for the comments the details that required corrections were modified.

- chapter 3, 'Discussion'- this chapter should be moved to the 'Introduction' section; this chapter should be about discussing the reported results; only the last paragraph is suitable for the 'Discussion';

Au: The section was re-written and it contains discussion of our findings as well as comparison to the literature.

- there is no 'Summary' or 'Conclusions' chapter in this article;

Au: We have added conclusions at the end of the discussion

- Figure captions in the main text should be short and refer only to what is shown in the specific figure;

Au: Figure captions were modified accordingly

- Results- this chapter should be divided into two subsections, i.e. I- concerns only the preparation of CaCO3 particles samples, II- concerns the influence of the silicon-containing additives;

Au: We thank the reviewer, subtitles were provided accordingly to clarify the results.

- Figure 1S- should be in the main text, not in the supplementary material;

Au: we are afraid that although there is relevant data related to the calcium chelator, it deviates from the main idea discussed and for that we maintain the figure in the supplementary section.

- p.2- lines 81-86- there is no reference to this fragment in the subsection 'preparation ..', where only the sodium silicate solution is mentioned; what exactly were these gels?; Figure 3S should be in the main text, not in the supplementary material; Figure 7- wrong numbering, please correct it;

Au: As mentioned in the text silicon gel and grease are known to contain various silicon-containing molecules but the exact compositions are not well defined, therefore this data was included in the SI and we included in the main text the data with a well-defined soluble silicate material. The numbering was fixed accordingly.

- Figure 3- the marks are not well visible on it, descriptions have been 'cutted'; please remove fragment A from the figure - it does not add anything significant to the text;

Au: The marks were corrected.

- Figures 5S and 6S- should be in the main text, not in the supplementary material.

Au: We afraid that inclusion of these two figures would deviate the main massage of the paper and we hope that in the re-written version would be clearer. In case there will be consent that this is critical of-course we may modify.

Reviewer 3 Report

This study deals with the polymorphism of calcium carbonate.  Control of crystalline states of calcium carbonate is very important for industrial and biotechnological applications.  In this paper, the authors synthesized calcium carbonate particles and investigated their morphologies.  The descriptions of the experimental data seem to be reasonable, but there are some insufficient points listed below.  Therefore, this paper can be accepted for publication in the molecules after minor revision.

(1) In the TEM measurements, particle size distribution must be shown.

(2) PXRD data must be analyzed.  Crystal parameters should be estimated from the obtained data.

(3) P.2  L.85  Fig.7 is missing.

Author Response

  • In the TEM measurements, particle size distribution must be shown.

Au: We consider TEM and SEM analysis as semi-quantitative methods to assess size of particles. Therefore we provided estimations and size ranges. Since we captured representative optic fields we prefer to show the information in a visual way rather than providing statistics which may not be accurate by image analysis.      

  • PXRD data must be analyzed.  Crystal parameters should be estimated from the obtained data.

Au: Thanks for the comment. We added further information (see paragraph line 199)

(3) P.2  L.85  Fig.7 is missing.

Au: Thanks for the comment. Figure numbers were corrected.

Round 2

Reviewer 2 Report

Authors of the article entitled 'Controlling Calcium Carbonate Particle Morphology, Size and Molecular Order Using Silicate' have addressed most of the previous comments, but the article still contains some minor errors:

- subsection 2.1.- it should be formula (singular) instead of formulae (plural);

- subsection 2.1- labels with the size of the packaging should be removed, for example: "Sigma, 31218, Germany" should be "Sigma, Germany" etc.;

- Figure captions are still too long- explanations should be in the main text, not in the figure caption, for example (Figure 2): "Au detected is due to sample coating prior to SEM" should be moved to the main text;

- Chapters Discussion, Conclusion, Acknowledgements have wrong numbering.

Author Response

We thank the reviewer for the comments. All of the mentioned comments were corrected.